# Certifiable Robustness to Graph Perturbations

**Aleksandar Bojchevski**
Technical University of Munich
`a.bojchevski@in.tum.de`

**Stephan Günnemann**
Technical University of Munich
`guennemann@in.tum.de`

## Abstract

Despite the exploding interest in graph neural networks there has been little effort to verify and improve their robustness. This is even more alarming given recent findings showing that they are extremely vulnerable to adversarial attacks on both the graph structure and the node attributes. We propose the first method for verifying certifiable (non-)robustness to graph perturbations for a general class of models that includes graph neural networks and label/feature propagation. By exploiting connections to PageRank and Markov decision processes our certificates can be efficiently (and under many threat models exactly) computed. Furthermore, we investigate robust training procedures that increase the number of certifiably robust nodes while maintaining or improving the clean predictive accuracy.

## 1   Introduction

As the number of machine learning models deployed in the real world grows, questions regarding their robustness become increasingly important. In particular, it is critical to assess their vulnerability to adversarial attacks – deliberate perturbations of the data designed to achieve a specific (malicious) goal. Graph-based models suffer from poor adversarial robustness [13, 60], yet in domains where they are often deployed (e.g. the Web) [50], adversaries are pervasive and attacks have a low cost [9, 26]. Even in scenarios where adversaries are not present such analysis is important since it allows us to reason about the behavior of our models in the worst case (i.e. treating nature as an adversary).

Here we focus on semi-supervised node classification – given a single large (attributed) graph and the class labels of a few nodes the goal is to predict the labels of the remaining unlabelled nodes. Graph Neural Networks (GNNs) have emerged as the de-facto way to tackle this task, significantly improving performance over the previous state-of-the-art. They are used for various high impact applications across many domains such as: protein interface prediction [20], classification of scientific papers [28], fraud detection [44], and breast cancer classification [36]. Therefore, it is crucial to asses their sensitivity to adversaries and ensure they behave as expected.

However, despite their popularity there is scarcely any work on certifying or improving the robustness of GNNs. As shown in Zügner et al. [60] node classification with GNNs is not robust and can even be attacked on multiple fronts – slight perturbations of either the node features or the graph structure can lead to wrong predictions. Moreover, since we are dealing with non i.i.d. data by taking the graph structure into account, robustifying GNNs is more difficult compared to traditional models – perturbing only a few edges affects the predictions for all nodes. What can we do to fortify GNNs and make sure they produce reliable predictions in the presence of adversarial perturbations?

We propose the first method for provable robustness regarding perturbations of the graph structure. Our approach is applicable to a general family of models where the predictions are a linear function of (personalized) PageRank. This family includes GNNs [29] and other graph-based models such as label/feature propagation [7, 53]. Specifically, we provide: **1. Certificates**: Given a trained model and a general set of admissible graph perturbations we can efficiently verify whether a node is certifiably robust – there exists no perturbation that can change its prediction. We also provide non-

robustness certificates via adversarial examples. **2. Robust training**: We investigate robust training schemes based on our certificates and show that they improve both robustness and clean accuracy. Our theoretical findings are empirically demonstrated and the code is provided for reproducibility[1]. Interestingly, in contrast to existing works on provable robustness [23, 46, 59] that derive bounds (by relaxing the problem), we can efficiently compute exact certificates for some threat models.

## 2 Related work

Neural networks [41, 21], and recently graph neural networks [13, 60, 58] and node embeddings [5] were shown to be highly sensitive to small adversarial perturbations. There exist many (heuristic) approaches aimed at robustifying these models, however, they have only limited usefulness since there is always a new attack able to break them, leading to a cat-and-mouse game between attackers and defenders. A more promising line of research studies certifiable robustness [23, 35, 46]. Certificates provide guarantees that no perturbation regarding a specific threat model will change the prediction of an instance. So far there has been almost no work on certifying graph-based models.

Different heuristics have been explored in the literature to improve robustness of graph-based models: (virtual) adversarial training [10, 16, 40, 49], trainable edge weights [48], graph encoder refining and adversarial contrastive learning [45], transfer learning [42], smoothing distillation [10], decoupling structure from attributes [31], measuring logit discrepancy [51], allocating reliable queries [56], representing nodes as Gaussian distributions [57], and Bayesian graph neural networks [52]. Other robustness aspects of graph-based models (e.g. noise or anomalies) have also been investigated [3, 6, 24]. However, none of these works provide provable guarantees or certificates.

Zügner & Günnemann [59] is the only work that proposes robustness certificates for graph neural networks (GNNs). However, their approach can handle perturbations only to the *node attributes*. Our approach is completely orthogonal to theirs since we consider adversarial perturbations to the *graph structure* instead. Furthermore, our certificates are also valid for other semi-supervised learning approaches such as label/feature propagation. Nonetheless, there is a critical need for both types of certificates given that GNNs are shown to be vulnerable to attacks on both the attributes and the structure. As future work, we aim to consider perturbations of the node features and the graph jointly.

## 3 Background and preliminaries

Let $G = (\mathcal{V}, \mathcal{E})$ be an attributed graph with $N = |\mathcal{V}|$ nodes and edge set $\mathcal{E} \subseteq \mathcal{V} \times \mathcal{V}$. We denote with $\boldsymbol{A} \in \{0, 1\}^{N \times N}$ the adjacency matrix and $\boldsymbol{X} \in \mathbb{R}^{N \times D}$ the matrix of $D$-dimensional node features for each node. Given a subset $\mathcal{V}_L \subseteq \mathcal{V} = \{1, \ldots, N\}$ of labelled nodes the goal of semi-supervised node classification is to predict for each node $v \in \mathcal{V}$ one class in $\mathcal{C} = \{1, \ldots, K\}$. We focus on deriving (exact) robustness certificates for graph neural networks via optimizing personalized PageRank. We also show (Appendix 8.1) how to apply our approach for label/feature propagation [7].

**Topic-sensitive PageRank.** The topic-sensitive PageRank [22, 27] vector $\boldsymbol{\pi}_G(\boldsymbol{z})$ for a graph $G$ and a probability distribution over nodes $\boldsymbol{z}$ is defined as $\boldsymbol{\pi}_{G,\alpha}(\boldsymbol{z}) = (1 - \alpha)(\boldsymbol{I}_N - \alpha \boldsymbol{D}^{-1}\boldsymbol{A})^{-1}\boldsymbol{z}$.[2] Here $\boldsymbol{D}$ is a diagonal matrix of node out-degrees with $\boldsymbol{D}_{ii} = \sum_j \boldsymbol{A}_{ij}$. Intuitively, $\boldsymbol{\pi}(\boldsymbol{z})_u$ represent the probability of random walker on the graph to land at node $u$ when it follows edges at random with probability $\alpha$ and teleports back to the node $v$ with probability $(1 - \alpha)\boldsymbol{z}_v$. Thus, we have $\boldsymbol{\pi}(\boldsymbol{z})_u \geq 0$ and $\sum_u \boldsymbol{\pi}(\boldsymbol{z})_u = 1$. For $\boldsymbol{z} = \boldsymbol{e}_v$, the $v$-th canonical basis vector, we get the personalized PageRank vector for node $v$. We drop the index on $G, \alpha$ and $\boldsymbol{z}$ in $\boldsymbol{\pi}_{G,\alpha}(\boldsymbol{z})$ when they are clear from the context.

**Graph neural networks.** As an instance of graph neural network (GNN) methods we consider an adaptation of the recently proposed PPNP approach [29] since it shows superior performance on the semi-supervised node classification task [19]. PPNP unlike message-passing GNNs decouples the feature transformation from the propagation. We have:

$$\boldsymbol{Y} = \text{softmax}\big(\boldsymbol{\Pi}^{\text{sym}}\boldsymbol{H}\big), \quad \boldsymbol{H}_{v,:} = f_\theta(\boldsymbol{X}_{v,:}), \quad \boldsymbol{\Pi}^{\text{sym}} = (1 - \alpha)(\boldsymbol{I}_N - \alpha \boldsymbol{D}^{-1/2}\boldsymbol{A}\boldsymbol{D}^{-1/2})^{-1} \quad (1)$$

where $\boldsymbol{I}_N$ is the identity, $\boldsymbol{\Pi}^{\text{sym}} \in \mathbb{R}^{N \times N}$ is a symmetric propagation matrix, $\boldsymbol{H} \in \mathbb{R}^{N \times C}$ collects the individual per-node logits, and $\boldsymbol{Y} \in \mathbb{R}^{N \times C}$ collects the final predictions after propagation. A

neural network $f_\theta$ outputs the logits $\boldsymbol{H}_{v,:}$ by processing the features $\boldsymbol{X}_{v,:}$ of every node $v$ independently. Multiplying them with $\boldsymbol{\Pi}^{\text{sym}}$ we obtain the diffused logits $\boldsymbol{H}^{\text{diff}} := \boldsymbol{\Pi}^{\text{sym}}\boldsymbol{H}$ which implicitly incorporate the graph structure and avoid the expensive multi-hop message-passing procedure.

To make PPNP more amenable to theoretical analysis we replace $\boldsymbol{\Pi}^{\text{sym}}$ with the personalized PageRank matrix $\boldsymbol{\Pi} = (1 - \alpha)(\boldsymbol{I}_N - \alpha\boldsymbol{D}^{-1}\boldsymbol{A})^{-1}$ which has a similar spectrum. Here each row $\boldsymbol{\Pi}_{v,:} = \boldsymbol{\pi}(\boldsymbol{e}_v)$ equals to the personalized PageRank vector of node $v$. This model which we denote as $\boldsymbol{\pi}$-PPNP has similar prediction performance to PPNP. We can see that the diffused logit after propagation for class $c$ of node $v$ is a linear function of its personalized PageRank score: $\boldsymbol{H}_{v,c}^{\text{diff}} = \boldsymbol{\pi}(\boldsymbol{e}_v)^T\boldsymbol{H}_{:,c}$, i.e. a weighted combination of the logits of all nodes for class $c$. Similarly, the margin $m_{c_1,c_2}(v) = \boldsymbol{H}_{v,c_1}^{\text{diff}} - \boldsymbol{H}_{v,c_2}^{\text{diff}} = \boldsymbol{\pi}(\boldsymbol{e}_v)^T(\boldsymbol{H}_{:,c_1} - \boldsymbol{H}_{:,c_2})$ defined as the difference in logits for node $v$ for two given classes $c_1$ and $c_2$ is also linear in $\boldsymbol{\pi}(\boldsymbol{e}_v)$. If $\min_c m_{y_v,c}(v) < 0$, where $y_v$ is the ground-truth label for $v$, the node is *misclassified* since the prediction equals $\arg\max_c \boldsymbol{H}_{v,c}^{\text{diff}}$.

# 4  Robustness certificates

## 4.1  Threat model, fragile edges, global and local budget

We investigate the scenario in which a subset of edges in a directed graph are "fragile", i.e. an attacker has control over them, or in general we are not certain whether these edges are present in the graph. Formally, we are given a set of fixed edges $\mathcal{E}_f \subseteq \mathcal{E}$ that cannot be modified (assumed to be reliable), and set of fragile edges $\mathcal{F} \subseteq (\mathcal{V} \times \mathcal{V}) \setminus \mathcal{E}_f$. For each fragile edge $(i,j) \in \mathcal{F}$ the attacker can decide whether to include it in the graph or exclude it from the graph, i.e. set $\boldsymbol{A}_{ij}$ to 1 or 0 respectively. For any subset of included $\mathcal{F}_+ \subseteq \mathcal{F}$ edges we can form the perturbed graph $\tilde{G} = (\mathcal{V}, \tilde{\mathcal{E}} := \mathcal{E}_f \cup \mathcal{F}_+)$. An *excluded* fragile edge $(i,j) \in \mathcal{F} \setminus \mathcal{F}_+$ is a non-edge in $\tilde{G}$. This formulation is general, since we can set $\mathcal{E}_f$ and $\mathcal{F}$ arbitrarily. For example, for our certificate scenario given an existing clean graph $G = (\mathcal{V}, \mathcal{E})$ we can set $\mathcal{E}_f = \mathcal{E}$ and $\mathcal{F} \subseteq \mathcal{V} \times \mathcal{V}$ which implies the attacker can only add new edges to obtain perturbed graphs $\tilde{G}$. Or we can set $\mathcal{E}_f = \emptyset$ and $\mathcal{F} = \mathcal{E}$ so that the attacker can only remove edges, and so on. There are $2^{|\mathcal{F}|}$ (exponential) number of valid configurations leading to different perturbed graphs which highlights that certificates are challenging for graph perturbations.

In reality, perturbing an edge is likely to incur some cost for the attacker. To capture this we introduce a *global budget*. The constraint $|\tilde{\mathcal{E}} \setminus \mathcal{E}| + |\mathcal{E} \setminus \tilde{\mathcal{E}}| \le B$ implies that the attacker can make at most $B$ perturbations. The first term equals to the number of newly added edges, and the second to the number of removed existing edges. Here, including an edge that already exists does not count towards the budget. This is only a design choice that depends on the application, and our method works in general. Furthermore, perturbing many edges for a single node might not be desirable, thus we also allow to limit the number of perturbations locally. Let $\mathcal{E}^v = \{(v,j) \in \mathcal{E}\}$ be the set of edges that share the same source node $v$. Then, the constraint $|\tilde{\mathcal{E}}^v \setminus \mathcal{E}^v| + |\mathcal{E}^v \setminus \tilde{\mathcal{E}}^v| \le b_v$ enforces a local budget $b_v$ for the node $v$. By setting $b_v = |\mathcal{F}^v|$ and $B = |\mathcal{F}|$ we can model an unconstrained attacker. Letting $\mathcal{P}(\mathcal{F})$ be the power set of $\mathcal{F}$, we define the set of admissible perturbed graphs:

$$\mathcal{Q}_\mathcal{F} = \{(\mathcal{V}, \tilde{\mathcal{E}} := \mathcal{E}_f \cup \mathcal{F}_+) \mid \mathcal{F}_+ \in \mathcal{P}(\mathcal{F}), |\tilde{\mathcal{E}} \setminus \mathcal{E}| + |\mathcal{E} \setminus \tilde{\mathcal{E}}| \le B, |\tilde{\mathcal{E}}^v \setminus \mathcal{E}^v| + |\mathcal{E}^v \setminus \tilde{\mathcal{E}}^v| \le b_v, \forall v\} \quad (2)$$

## 4.2  Robustness certificates

**Problem 1.** *Given a graph $G$, a set of fixed $\mathcal{E}_f$ and fragile $\mathcal{F}$ edges, global $B$ and local $b_v$ budgets, target node $t$, and a model with logits $\boldsymbol{H}$. Let $y_t$ denote the class of node $t$ (predicted or ground-truth). The worst-case margin between class $y_t$ and class $c$ under any admissible perturbation $\tilde{G} \in \mathcal{Q}_\mathcal{F}$ is:*

$$m_{y_t,c}^*(t) = \min_{\tilde{G} \in \mathcal{Q}_\mathcal{F}} m_{y_t,c}(t) = \min_{\tilde{G} \in \mathcal{Q}_\mathcal{F}} \boldsymbol{\pi}_{\tilde{G}}(\boldsymbol{e}_t)^T (\boldsymbol{H}_{:,y_t} - \boldsymbol{H}_{:,c}) \quad (3)$$

*If $m_{y_t,*}^*(t) = \min_{c \ne y_t} m_{y_t,c}^*(t) > 0$, node $t$ is certifiably robust w.r.t. the logits $\boldsymbol{H}$, and the set $\mathcal{Q}_\mathcal{F}$.*

Our goal is to verify whether no admissible $\tilde{G} \in \mathcal{Q}_\mathcal{F}$ can change the prediction for a target node $t$. From Problem 1 we see that if the worst margin over all classes $m_{y_t,*}^*(t)$ is positive, then $m_{y_t,c}^*(t) > 0$, for all $y_t \ne c$, which implies that there exists *no* adversarial example within $\mathcal{Q}_\mathcal{F}$ that leads to a change in the prediction to some other class $c$, that is, the logit for the given class $y_t$ is always largest.

**Challenges and core idea.** From a cursory look at Eq. 3 it appears that finding the minimum is intractable. After all, our domain is discrete and we are optimizing over exponentially many configurations. Moreover, the margin is a function of the personalized PageRank which has a non-trivial dependency on the perturbed graph. But there is hope: For a fixed $\boldsymbol{H}$, the margin $m_{y_t,c}(t)$ is a linear function of $\boldsymbol{\pi}(\boldsymbol{e}_t)$. Thus, Problem 1 reduces to optimizing a linear function of personalized PageRank over a specific constraint set. This is the core idea of our approach. As we will show, if we consider only local budget constraints the exact certificate can be efficiently computed. This is in contrast to most certificates for neural networks that rely on different relaxations to make the problem tractable. Including the global budget constraint, however, makes the problem hard. For this case we derive an efficient to compute lower bound on the worst-case margin. Thus, if the lower bound is positive we can still guarantee that our classifier is robust w.r.t. the set of admissible perturbations.

## 4.3 Optimizing topics-sensitive PageRank with global and local constraints

We are interested in optimizing a linear function of the topic-sensitive PageRank vector of a graph by modifying its structure. That is, we want to configure a set of fragile edges into included/excluded to obtain a perturbed graph $\tilde{G}$ maximizing the objective. Formally, we study the general problem:

**Problem 2.** *Given a graph $G$, a set of admissible perturbations $\mathcal{Q}_{\mathcal{F}}$ as in Problem 1, and any fixed $\boldsymbol{z}, \boldsymbol{r} \in \mathbb{R}^N, \alpha \in (0,1)$ solve the following optimization problem:* $\max_{\tilde{G} \in \mathcal{Q}_{\mathcal{F}}} \boldsymbol{r}^T \boldsymbol{\pi}_{\tilde{G},\alpha}(\boldsymbol{z})$.

Setting $\boldsymbol{r} = -(\boldsymbol{H}_{:,y_t} - \boldsymbol{H}_{:,c})$ and $\boldsymbol{z} = \boldsymbol{e}_t$, we see that Problem 1 is a special case of Problem 2. We can think of $\boldsymbol{r}$ as a reward/cost vector, i.e. $\boldsymbol{r}_v$ is the reward that a random walker obtains when visiting node $v$. The objective value $\boldsymbol{r}^T \boldsymbol{\pi}(\boldsymbol{z})$ is proportional to the overall reward obtained during an infinite random walk with teleportation since $\boldsymbol{\pi}(\boldsymbol{z})_v$ exactly equals to the frequency of visits to $v$.

Variations and special cases of this problem have been previously studied [2, 11, 12, 15, 18, 25, 32]. Notably, Fercoq et al. [18] cast the problem as an average cost infinite horizon Markov decision process (MDP), also called ergodic control problem, where each node corresponds to a state and the actions correspond to choosing a subset of included fragile edges, i.e. we have $2^{|\mathcal{F}^v|}$ actions at each state $v$ (see also Fig. 2a). They show that despite the exponential number of actions, the problem can be efficiently solved in polynomial time, and they derive a value iteration algorithm with different local constraints. They enforce that the final perturbed graph has at most $b_v$ total number of edges per node, while we enforce that at most $b_v$ edges per node are perturbed (see Sec. 4.1).

**Our approach for local budget only.** Inspired by the MDP idea we derive a policy iteration (PI) algorithm which also runs in polynomial time [25]. Intuitively, every policy corresponds to a perturbed graph in $\mathcal{Q}_{\mathcal{F}}$, and each iteration improves the policy. The PI algorithm allows us to: incorporate our local constraints easily, take advantage of efficient solvers for sparse systems of linear equations (line 3 in Alg. 1), and implement the policy improvement step in parallel (lines 4-6 in Alg. 1). It can easily handle very large sets of fragile edges and it scales to large graphs.

**Proposition 1.** *Algorithm 1 which greedily selects the fragile edges finds an optimal solution for Problem 2 with only local constraints in a number of steps independent of the size of the graph.*

---

**Algorithm 1** POLICY ITERATION WITH LOCAL BUDGET

---

**Require:** Graph $G = (\mathcal{V}, \mathcal{E})$, reward $\boldsymbol{r}$, set of fixed $\mathcal{E}_f$ and fragile $\mathcal{F}$ edges, local budgets $b_v$
  1: Initialization: $\mathcal{W}_0 \subseteq \mathcal{F}$ as any arbitrary subset, $\boldsymbol{A}^G$ corresponding to $G$
  2: **while** $\mathcal{W}_k \neq \mathcal{W}_{k-1}$ **do**
  3:    Solve $(\boldsymbol{I}_N - \alpha \boldsymbol{D}^{-1}\boldsymbol{A})\boldsymbol{x} = \boldsymbol{r}$ for $\boldsymbol{x}$, where $\boldsymbol{A}_{ij} = 1 - \boldsymbol{A}_{ij}^G$ if $(i,j) \in \mathcal{W}_k$    # flip the edges
  4:    Let $l_{ij} \leftarrow (1 - 2\boldsymbol{A}_{ij}^G)(\boldsymbol{x}_j - \frac{\boldsymbol{x}_i - \boldsymbol{r}_i}{\alpha})$ for all $(i,j) \in \mathcal{F}$        # calculate the improvement
  5:    Let $\mathcal{L}_v \leftarrow \{(v,j) \in \mathcal{F} \mid l_{vj} > 0 \wedge l_{vj} \geq \text{top } b_v \text{ largest } l_{vj}\}, \forall v \in \mathcal{V}$
  6:    $\mathcal{W}_k \leftarrow \bigcup_v \mathcal{L}_v, \quad k \leftarrow k+1$
  7: **end while**
  8: **return** $\mathcal{W}_k$                # optimal graph $\tilde{G} \in \mathcal{Q}_{\mathcal{F}}$ obtained by flipping all $(i,j) \in \mathcal{W}_k$ of $G$

---

We provide the proof in Sec. 8.3 in the appendix. The main idea for Alg. 1 is starting from a random policy, in each iteration we first compute the mean reward before teleportation $\boldsymbol{x}$ for the current policy (line 3), and then greedily select the top $b_v$ edges that improve the policy (lines 4-6). This algorithm is guaranteed to converge to the optimal policy, and thus to the optimal configuration of fragile edges.

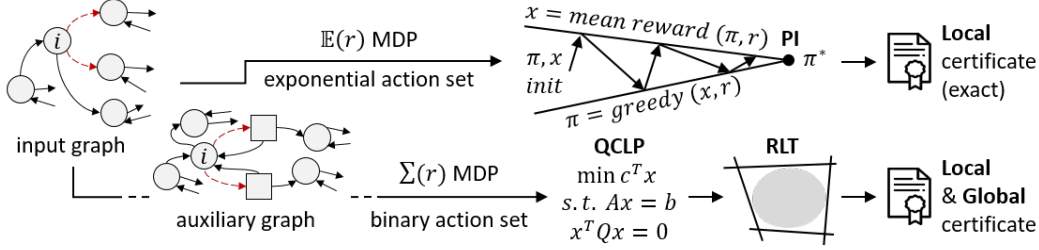

Figure 1: The upper part outlines our approach for local budget only: the exact certificate is efficiently computed with policy iteration. The lower part outlines our 3 step approach for local and global budget: (a) formulate an MDP on an auxiliary graph, (b) augment the corresponding LP with quadratic constraints to enforce the global budget, and (c) apply the RLT relaxation to the resulting QCLP.

**Certificate for local budget only.** Proposition 1 implies that for local constraints only, the optimal solution does not depend on the teleport vector $\boldsymbol{z}$. Regardless of the node $t$ (i.e. which $\boldsymbol{z} = \boldsymbol{e}_t$ in Eq. 3), the optimal edges to perturb are the same if the admissible set $\mathcal{Q}_{\mathcal{F}}$ and the reward $\boldsymbol{r}$ are the same. This means that for a fixed $\mathcal{Q}_{\mathcal{F}}$ we only need to run the algorithm $K \times K$ times to obtain the certificates *for all* $N$ nodes: For each pair of classes $c_1, c_2$ we have a different reward vector $\boldsymbol{r} = -(\boldsymbol{H}_{:,c_1} - \boldsymbol{H}_{:,c_2})$, and we can recover the *exact* worst-case margins $m^*_{y_t,*}(\cdot)$ for all $N$ nodes by just computing $\boldsymbol{\Pi}$ on the resulting $K \times K$ many perturbed graphs $\tilde{G}$. Now, $m^*_{y_t,*}(\cdot) > 0$ implies certifiable robustness, while $m^*_{y_t,*}(\cdot) < 0$ implies certifiable *non-robustness* due to the exactness of our certificate, i.e. we have found an adversarial example for node $t$.

**Our approach for both local and global budget.** Algorithm 1 cannot handle a global budget constraint, and in general solving Problem 2 with global budget is NP-hard. More specifically, it generalizes the Link Building problem [32] – find the set of $k$ optimal edges that point to a given node such that its PageRank score is maximized – which is W[1]-hard and for which there exists no fully-polynomial time approximation scheme (FPTAS). It follows that Problem 2 is also W[1]-hard and allows no FPTAS. We provide the proof and more details in Sec. 8.5 in the appendix. Therefore, we develop an alternative approach that consists of three steps and is outlined in the lower part of Fig. 1: (a) We propose an alternative unconstrained MDP based on an auxiliary graph which reduces the action set from exponential to binary by adding only $|\mathcal{F}|$ auxiliary nodes; (b) We reformulate the problem as a non-convex Quadratically Constrained Linear Program (QCLP) to be able to handle the global budget; (c) We utilize the Reformulation Linearization Technique (RLT) to construct a convex relaxation of the QCLP, enabling us to efficiently compute a lower bound on the worst-case margin.

**(a) Auxiliary graph.** Given an input graph we add one auxiliary node $v_{ij}$ for each fragile edge $(i, j) \in \mathcal{F}$. We define a total cost infinite horizon MDP on this auxiliary graph (Fig. 2b) that solves Problem 2 *without* constraints. The MDP is defined by the 4-tuple $(\mathcal{S}, (\mathcal{A}_i)_{i \in \mathcal{S}}, p, r)$, where $\mathcal{S}$ is the state space (preexisting and auxiliary nodes), and $\mathcal{A}_i$ is the set of admissible actions in state $i$. Given action $a \in \mathcal{A}_i$, $p(j|i, a)$ is the probability to go to state $j$ from state $i$ and $r(i, a)$ the instantaneous reward. Each *preexisting* node $i$ has a single action $\mathcal{A}_i = \{a\}$, reward $r(i, a) = \boldsymbol{r}_i$, and uniform transitions $p(v_{ij}|i, a) = d_i^{-1}, \forall v_{ij} \in \mathcal{F}^i$, discounted by $\alpha$ for the fixed edges $p(j|i, a) = \alpha \cdot d_i^{-1}, \forall (i, j) \in \mathcal{E}_f$, where $d_i = |\mathcal{E}_f^i \cup \mathcal{F}^i|$ is the degree. For each *auxiliary* node we allow two actions $\mathcal{A}_{v_{ij}} = \{\text{on}, \text{off}\}$. For action "off" node $v_{ij}$ goes back to node $i$ with probability 1 and obtains reward $-\boldsymbol{r}_i$: $p(i|v_{ij}, \text{off}) = 1, r(v_{ij}, \text{off}) = -\boldsymbol{r}_i$. For action "on" node $v_{ij}$ goes only to node $j$ with probability $\alpha$ (the model is substochastic) and obtains 0 reward: $p(j|v_{ij}, \text{on}) = \alpha, r(v_{ij}, \text{on}) = 0$. We introduce fewer aux. nodes compared to previous work [11, 17].

**(b) Global and local budgets QCLP.** Based on this unconstrained MDP, we can derive a corresponding linear program (LP) solving the same problem [34]. Since the MDP on the auxiliary graph has (at most) binary action sets, the LP has only $2|\mathcal{V}| + 3|\mathcal{F}|$ constraints and variables. This is in strong contrast to the LP corresponding to the previous average cost MPD [18] operating directly on the original graph that has an exponential number of constraints and variables. Lastly, we enrich the LP for the aux. graph MDP with additional constraints enforcing the local and global budgets. The constraints for the local budget are linear, however, the global budget requires quadratic constraints resulting in a quadratically constrained linear program (QCLP) that exactly solves Problem 2.

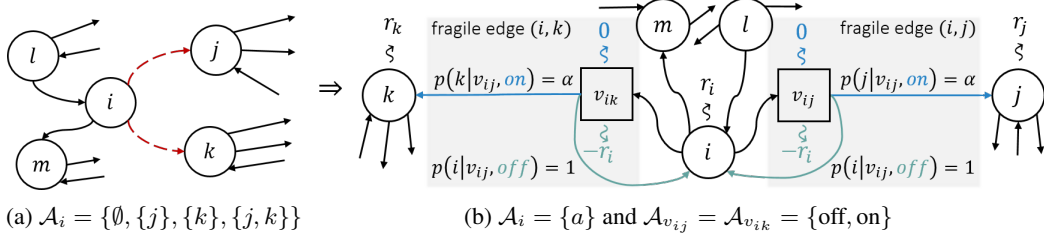

(a) $\mathcal{A}_i = \{\emptyset, \{j\}, \{k\}, \{j, k\}\}$        (b) $\mathcal{A}_i = \{a\}$ and $\mathcal{A}_{v_{ij}} = \mathcal{A}_{v_{ik}} = \{\text{off, on}\}$

Figure 2: Construction of the auxiliary graph. For each fragile edge $(i, j)$ marked with a red dashed line, we add one node $v_{ij}$ and two actions: {on, off} to the auxiliary graph. If the edge is configured as "on" $v_{ij}$ goes back to node $i$ with prob. 1. If configured as "off" it goes only to node $j$ with prob. $\alpha$.

**Proposition 2.** *Solving the following QCLP (with decision variables $x_v, x_{ij}^0, x_{ij}^1, \beta_{ij}^0, \beta_{ij}^1$) is equivalent to solving Problem 2 with local and global constraints, i.e. the value of the objective function is the same in the optimal solution. We can recover $\boldsymbol{\pi}(z)_v$ from $x_v$ via $\boldsymbol{\pi}(z)_v = (1 - k_v d_v^{-1}) x_v$. Here $k_v$ is the number of "off" fragile edges (the ones where $x_{ij}^0 > 0$) in the optimal solution.*

$$\max \sum_{v \in \mathcal{V}} x_v \boldsymbol{r}_v - \sum_{(i,j) \in \mathcal{F}} x_{ij}^0 \boldsymbol{r}_i \tag{4a}$$

$$x_v - \alpha \underbrace{\sum_{(i,v) \in \mathcal{E}_f} \frac{x_i}{d_i}}_{\text{incoming fixed edges}} - \alpha \underbrace{\sum_{(j,v) \in \mathcal{F}} x_{jv}^1}_{\text{incoming "on" edges}} - \underbrace{\sum_{(v,k) \in \mathcal{F}} x_{vk}^0}_{\text{returning "off" edges}} = (1 - \alpha) \boldsymbol{z}_v \quad \forall v \in \mathcal{V} \tag{4b}$$

$$x_{ij}^0 + x_{ij}^1 = \frac{x_i}{d_i}, \qquad x_{ij}^0 \geq 0, \qquad x_{ij}^1 \geq 0 \qquad\qquad\qquad \forall (i,j) \in \mathcal{F} \tag{4c}$$

$$\sum_{(v,i) \in \mathcal{F}} \underbrace{[(v,i) \in \mathcal{E}] x_{ij}^0}_{\text{removed existing edges}} + \underbrace{[(v,i) \notin \mathcal{E}] x_{ij}^1}_{\text{newly added edges}} \leq \frac{x_v}{d_v} b_v, \qquad x_v \geq 0 \qquad \forall v \in \mathcal{V} \tag{4d}$$

$$x_{ij}^0 \beta_{ij}^1 = 0, \quad x_{ij}^1 \beta_{ij}^0 = 0, \quad \beta_{ij}^1 = 1 - \beta_{ij}^0, \quad 0 \leq \beta_{ij}^0 \leq 1 \qquad \forall (i,j) \in \mathcal{F} \tag{4e}$$

$$\sum_{(i,j) \in \mathcal{F}} [(i,j) \in \mathcal{E}] \beta_{ij}^0 + [(i,j) \notin \mathcal{E}] \beta_{ij}^1 \leq B \tag{4f}$$

**Key idea and insights.** Eqs. 4b and 4c correspond to the LP of the unconstrained MDP. Intuitively, the variable $x_v$ maps to the PageRank score of node $v$, and from the variables $x_{ij}^0 / x_{ij}^1$ we can recover the optimal policy: if the variable $x_{ij}^0$ (respectively $x_{ij}^1$) is non-zero then in the optimal policy the fragile edge $(i, j)$ is turned off (respectively on). Since there exists a deterministic optimal policy, only one of them is non-zero but never both. Eq. 4d corresponds to the local budget. Remarkably, despite the variables $x_{ij}^0 / x_{ij}^1$ not being integral, since they share the factor $x_i d_i^{-1}$ from Eq. 4c we can exactly count the number of edges that are turned off or on using only linear constraints. Eqs. 4e and 4f enforce the global budget. From Eq. 4e we have that whenever $x_{ij}^0$ is nonzero it follows that $\beta_{ij}^1 = 0$ and $\beta_{ij}^0 = 1$ since that is the only configuration that satisfies the constraints (similarly for $x_{ij}^1$). Intuitively, this effectively makes the $\beta_{ij}^0 / \beta_{ij}^1$ variables "counters" and we can utilize them in Eq. 4f to enforce the total number of perturbed edges to not exceed $B$. See detailed proof in Sec. 8.3.

**(c) Efficient Reformulation Linearization Technique (RLT).** The quadratic constraints in our QCLP make the problem non-convex and difficult to solve. We relax the problem using the Reformulation Linearization Technique (RLT) [38] which gives us an upper bound on the objective. The alternative SDP-relaxation [43] based on semidefinite programming is not suitable for our problem since the constraints are trivially satisfied (see Appendix 8.4 for details). While in general, the RLT introduces many new variables (replacing each product term $m_i m_j$ with a variable $M_{ij}$) along with multiple new linear inequality constraints, it turns out that in our case the solution is highly compact:

**Proposition 3.** *Given a fixed upper bound $\overline{x_v}$ for $x_v$ and using the RLT relaxation, the quadratic constraints in Eqs. 4e and 4f transform into the following single linear constraint.*

$$\sum_{(i,j) \in \mathcal{F}} [(i,j) \in \mathcal{E}] x_{ij}^0 d_i (\overline{x_i})^{-1} + [(i,j) \notin \mathcal{E}] x_{ij}^1 d_i (\overline{x_i})^{-1} \leq B \tag{5}$$

Proof provided in Sec. 8.3 in the appendix. By replacing Eqs. 4e and 4f with Eq. 5 in Proposition 2, we obtain a linear program which can be efficiently solved. Remarkably, we only have $x_v, x_{ij}^0, x_{ij}^1$ as decision variables since we were able to eliminate all other variables. The solution is an upper bound on the solution for Problem 2 and a lower bound on the solution for Problem 1. The final relaxed QCLP can also be interpreted as a constrained MPD with a single additional constraint (Eq. 5) which admits a possibly randomized optimal policy with at most one randomized state [1].

**Certificate for local and global budget.** To solve the relaxed QCLP and compute the final certificate we need to provide the upper bounds $\overline{x_v}$ for the constraint in Eq. 5. Since the quality of the RLT relaxation depends on the tightness of these upper bounds, we have to carefully select them. We provide here one solution (see Sec. 8.6 in the appendix for a faster to compute, but less tight, alternative): Given an instance of Problem 2, we can set the reward to $\boldsymbol{r} = \boldsymbol{e}_v$ and invoke Algorithm 1, which is highly efficient, using the same fragile set and the same local budget. Since this explicitly maximizes $x_v$, the objective value of the problem is guaranteed to give a valid upper bound $\overline{x}_v$. Invoking this procedure for every node, leads to the required upper bounds.

Now, to compute the certificate with local and global budget for a target node $t$, we solve the relaxed problem for all $c \neq y_t$, leading to objective function values $L_{ct} \geq -m_{y_t,c}^*(t)$ (minus due to the change from min to max). Thus, $L_{*,t} = \min_{c \neq y_t} -L_{ct}$ is a lower bound on the worst-case margin $m_{y_t,*}^*(t)$. If the lower bound is positive then node $t$ is guaranteed to be certifiably robust – there exists no adversarial attack (among all graphs in $\mathcal{Q}_\mathcal{F}$) that can change the prediction for node $t$.

For our policy iteration approach if $m_{y_t,*}^*(t) < 0$ we are guaranteed to have found an adversarial example since the certificate is exact, i.e. we also have a non-robustness certificate. However in this case, if the lower bound $L_{*,t}$ is negative we do not necessarily have an adversarial example. Instead, we can perturb the graph with the optimal configuration of fragile edges for the relaxed problem, and inspect whether the predictions change. See Fig.1 for an overview of both approaches.

# 5 Robust training for graph neural networks

In Sec. 4 we introduced two methods to efficiently compute certificates given a trained $\boldsymbol{\pi}$-PPNP model. We now show that these can naturally be used to go one step further – to *improve* the robustness of the model. The main idea is to utilize the worst-case margin during training to encourage the model to learn more robust weights. Optimizing some robust loss $\mathcal{L}_\theta$ with respect to the model parameters $\theta$ (e.g. for $\boldsymbol{\pi}$-PNPP $\theta$ are the neural network parameters) that depends on the worst-case margin $m_{y_v,*}^*(v)$ is generally hard since it involves an inner optimization problem, namely finding the worst-case margin. This prevents us to easily take the gradient of $m_{y_v,c}^*(v)$ (and, thus, $\mathcal{L}_\theta$) w.r.t. the parameters $\theta$. Previous approaches tackle this challenge by using the dual [46].

Inspecting our problem, however, we see that we can directly compute the gradient. Since $m_{y_v,c}^*(v)$ (respectively the corresponding lower bound) is a linear function of $\boldsymbol{H} = f_\theta(\boldsymbol{X})$ and $\boldsymbol{\pi}_G$, and furthermore the admissible set $\mathcal{Q}_\mathcal{F}$ over which we are optimizing is compact, it follows from Danskin's theorem [14] that we can simply compute the gradient of the loss at the optimal point. We have $\frac{\partial m_{y_v,c}^*(v)}{\partial \boldsymbol{H}_{i,y_v}} = \boldsymbol{\pi}^*(\boldsymbol{e}_v)_i$ and $\frac{\partial m_{y_v,c}^*(v)}{\partial \boldsymbol{H}_{i,c}} = -\boldsymbol{\pi}^*(\boldsymbol{e}_v)_i$, i.e. the gradient equals to the optimal ($\pm$) PageRank scores computed in our certification approaches.

**Robust training.** To improve robustness Wong & Kolter [46] proposed to optimize the *robust cross-entropy loss*: $\mathcal{L}_{RCE} = \mathcal{L}_{CE}(y_v^*, -\boldsymbol{m}_{y_v}^*(v))$, where $\mathcal{L}_{CE}$ is the standard cross-entropy loss operating on the logits, and $\boldsymbol{m}_{y_v}^*(v)$ is a vector such that at index $c$ we have $m_{y_v,c}^*(v)$. Previous work has shown that if the model is overconfident there is a potential issue when using $\mathcal{L}_{RCE}$ since it encourages high certainty under the worst-case perturbations [58]. Therefore, we also study the alternative *robust hinge loss*. Since the attacker wants to minimize the worst-case margin $m_{y_t,*}^*(t)$ (or its lower bound), a straightforward idea is to try to maximize it during training. To achieve this we add a hinge loss penalty term to the standard cross-entropy loss. Specifically: $\mathcal{L}_{CEM} = \sum_{v \in \mathcal{V}_L} \left[ \mathcal{L}_{CE}(y_v^*, \boldsymbol{H}_{v,:}^{\text{diff}}) + \sum_{c \in \mathcal{C}, c \neq y_v^*} \max(0, M - m_{y_v,c}^*(v)) \right]$. The second term for a single node $v$ is positive if $m_{y_v,c}^*(v) < M$ and zero otherwise – the node $v$ is certifiably robust with a margin of at least $M$. Effectively, if all training nodes are robust, the second term becomes zero, thus, reducing $\mathcal{L}_{CEM}$ to the standard cross-entropy loss with robustness guarantees. Note again that we can easily compute the gradient of these losses w.r.t. the (neural network) parameters $\theta$.

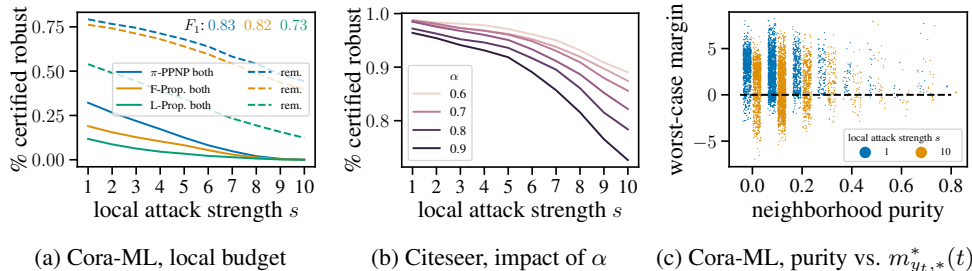

(a) Cora-ML, local budget     (b) Citeseer, impact of $\alpha$     (c) Cora-ML, purity vs. $m^*_{y_t,*}(t)$

Figure 3: Increasing local attack strength $s$ (local budget $b_v = \max(d_v - 11 + s, 0)$) decreases ratio of certified nodes. (a) The graph is more robust to removing edges, $\boldsymbol{\pi}$-PPNP is most robust overall. (b) Lowering $\alpha$ improves the robustness. (c) Nodes with higher neighborhood purity are more robust.

## 6 Experimental results

**Setup.** We focus on evaluating the robustness of $\boldsymbol{\pi}$-PPNP without robust training and label/feature propagation using our two certification methods. We also verify that robust training improves the robustness of $\boldsymbol{\pi}$-PPNP while maintaining high predictive accuracy. We demonstrate our claims on two publicly available datasets: Cora-ML ($N = 2,995, |\mathcal{E}| = 8,416, D = 2,879, K = 7$) [4, 30] and Citeseer ($N = 3,312, |\mathcal{E}| = 4,715, D = 3,703, K = 6$) [37] with further experiments on Pubmed ($N = 19,717, |\mathcal{E}| = 44,324, D = 500, K = 3$) [37] in the appendix. We configure $\boldsymbol{\pi}$-PPNP with one hidden layer of size 64 and set $\alpha = 0.85$. We select 20 nodes per class for the train/validation set and use the rest for the test set. We compute the certificates w.r.t. the predictions, i.e. we set $y_t$ in $m^*_{y_t,*}(t)$ to the predicted class for node $t$ on the clean graph. See Sec. 8.2 in the appendix for further experiments and Sec. 8.7 for more details. Note, we do not need to compare to any previously introduced adversarial attacks on graphs [13, 58, 60], since by the definition of a certificate for a certifiably robust node w.r.t. a given admissible set $\mathcal{Q}_\mathcal{F}$ there exist no successful attack within that set.

We construct several different configurations of fixed and fragile edges to gain a better understanding of the robustness of the methods to different kind of adversarial perturbations. Namely, "both" refers to the scenario where $\mathcal{F} = \mathcal{V} \times \mathcal{V}$, i.e. the attacker is allowed to add or remove *any* edge in the graph, while "rem." refers to the scenario where $\mathcal{F} = \mathcal{E}$ for a given graph $G = (\mathcal{V}, \mathcal{E})$, i.e. the attacker can only remove existing edges. In addition, for all scenarios we specify the fixed set as $\mathcal{E}_f = \mathcal{E}_{mst}$, where $(i,j) \in \mathcal{E}_{mst}$ if $(i,j)$ belongs to the minimum spanning tree (MST) on the graph $G$.[3]

**Robustness certificates: Local budget only.** We investigate the robustness of different graphs and semi-supervised node classification methods when the attacker has only local budget constraints. We set the local budget $b_v = \max(d_v - 11 + s, 0)$ relative to the degree $d_v$ of node $v$ in the clean graph, and we vary the *local attack strength $s$* with lower $s$ leading to a more restrictive budget. Such relative budget is justified since higher degree nodes tend to be more robust in general [59, 60]. We then apply our policy iteration algorithm to compute the (exact) worst-case margin for each node.

In Fig. 3a we see that the number of certifiably robust nodes when the attacker can only remove edges is significantly higher compared to when they can also add edges which is consistent with previous work on adversarial attacks [60]. As expected, the share of robust nodes decreases with higher budget, and $\boldsymbol{\pi}$-PPNP is significantly more robust than label propagation since besides the graph it also takes advantage of the node attributes. Feature propagation has similar performance ($F_1$ score) but it is less robust. Note that since our certificate is exact, the remaining nodes are certifiably non-robust! In Sec. 8.2 in the appendix we also investigate certifiable accuracy – the ratio of nodes that are both certifiably robust and at the same time have a correct prediction. We find that the certifiable accuracy is relatively close to the clean accuracy, and it decreases gracefully as we in increase the budget.

**Analyzing influence on robustness.** In Fig. 3b we see that decreasing the damping factor $\alpha$ is an effective strategy to significantly increase the robustness with no noticeable loss in accuracy (at most 0.5% for any $\alpha$, not shown). Thus, $\alpha$ provides a useful trade-off between robustness and the size of the effective neighborhood: higher $\alpha$ implies higher PageRank scores (i.e. higher influence) for the

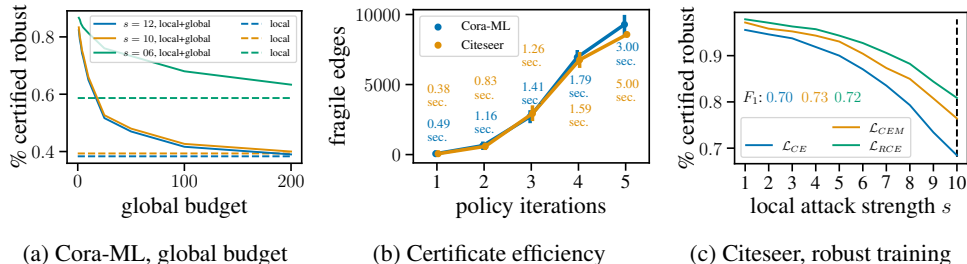

(a) Cora-ML, global budget   (b) Certificate efficiency   (c) Citeseer, robust training

Figure 4: (a) The global budget can significantly restrict the attacker compared to having only local constraints. (b) Even for large fragile sets Algorithm 1 only needs few iterations to find the optimal PageRank. (c) Our robust training successfully increases the percentage of certifiably robust nodes. The increase is largest for the local attack strength that we used during training ($s = 10$, dashed line).

neighbors. In general we recommend to set the value as low as the accuracy allows. In Fig. 3c we investigate what contributes to certain nodes being more robust than others. We see that neighborhood purity – the share of nodes with the same class in a respective node's two-hop neighborhood – plays an important role. High purity leads to high worst-case margin, which translates to certifiable robustness.

**Robustness certificates: Local and global budget.** We demonstrate our second approach based on the relaxed QCLP problem by analyzing the robustness as we increase the global budget. We set $\mathcal{F} = \mathcal{E}$, i.e. the attacker can only remove edges, and vary the local attack strength $s$ corresponding to local budget $b_v = \max(d_v - 11 + s, 0)$. We see in Fig.4a that by additionally enforcing a global budget we can significantly restrict the success of the attacker compared to having only a local budget (dashed lines). The global constraint increases the number of robust nodes, validating our approach.

**Efficiency.** Fig. 4b demonstrates the efficiency of our approach: even for fragile sets as large as $10^4$, Algorithm 1 finds the optimal solution in just a few iterations. Since each iteration is itself efficient by utilizing sparse matrix operations, the overall wall clock runtime (shown as text annotation) is on the order of few seconds. In Sec. 8.2 in the appendix, we further investigate the runtime as we increase the number of nodes in the graph, as well as the runtime of our relaxed QCLP.

**Robust training.** While not being our core focus, we investigate whether robust training improves the certifiable robustness of GNNs. We set the fragile set $\mathcal{F} = \mathcal{E}$ and vary the local budget. The vertical line on Fig. 4c indicates the local budget used to train the robust models with losses $\mathcal{L}_{RCE}$ and $\mathcal{L}_{CEM}$. We see that both of our approaches are able to improve the percent of certifiably robust nodes, with the largest improvement (around 13% increase) for the local attack strength we trained on ($s = 10$). Furthermore, the $F_1$ scores on the test split for Citeseer are as follows: 0.70 for $\mathcal{L}_{CE}$, 0.72 for $\mathcal{L}_{RCE}$, and 0.73 for $\mathcal{L}_{CEM}$, i.e. the robust training besides improving the ratio of certified nodes, it also improves the clean predictive accuracy of the model. $\mathcal{L}_{RCE}$ has a higher certifiable robustness, but $\mathcal{L}_{CEM}$ has a higher $F_1$ score. There is room for improvement in how we approach the robust training: e.g. similar to Zügner & Günnemann [59] we can optimize over the worst-case margin for the unlabeled in addition to the labeled nodes. We leave this as a future research direction.

## 7   Conclusion

We derive the first (non-)robustness certificate for graph neural networks regarding perturbations of the graph structure, and the first certificate overall for label/feature propagation. Our certificates are flexible w.r.t. the threat model, can handle both local (per node) and global budgets, and can be efficiently computed. We also propose a robust training procedure that increases the number of certifiably robust nodes while improving the predictive accuracy. As future work, we aim to consider perturbations and robustification of the node features and the graph structure jointly.

### Acknowledgments

This research was supported by the German Research Foundation, Emmy Noether grant GU 1409/2-1, and the German Federal Ministry of Education and Research (BMBF), grant no. 01IS18036B. The authors of this work take full responsibilities for its content.

## Footnotes

[1]Code, data, and supplementary material available at `https://www.daml.in.tum.de/graph-cert`

[2]In practice we do not invert the matrix, but rather we solve the associated sparse linear system of equations.

[3] Fixing the MST edges ensures that every node is reachable by every other node for any policy. This is only to simplify our earlier exposition regarding the MDPs and can be relaxed to e.g. reachable at the optimal policy.

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
