[Supplementary Material]

# 8 Appendix for "Certifiable Robustness to Graph Perturbations"

## 8.1 Certificates for Label Propagation and Feature Propagation

Label propagation is a classic method for semi-supervised node classification, and there have been many variants proposed over the year [54, 55, 53]. The general idea is to find a classification function $\boldsymbol{F}$ such that the training nodes are predicted correctly and the predicted labels change smoothly over the graph. We can express this formally via the following optimization problem [39]:

$$\min_{\boldsymbol{F}} \left\{ \sum_{i=1}^{N} \sum_{j=1}^{N} \boldsymbol{A}_{ij} \left\| d_i^{\sigma-1} F_{i*} - d_j^{\sigma-1} F_{j*} \right\|^2 + \mu \sum_{i=1}^{N} d_i^{2\sigma-1} \left\| F_{i*} - H_{i*} \right\|^2 \right\} \tag{6}$$

where, $d_i$ is the node degree, $\mu$ is a regularization parameter trading off smoothness and predicting the labeled nodes correctly, $\sigma$ is a hyper-parameter, and $\boldsymbol{H}$ is a matrix where the rows are one-hot vectors for the training nodes and zero vectors otherwise (i.e. $\boldsymbol{H}_{vc} = 1$ if $\{v \in \mathcal{V}_L \ \wedge \ y_v = c\}$ and $\boldsymbol{H}_{vc} = 0$ otherwise). The resulting matrix $\boldsymbol{F} \in \mathbb{R}^{N \times K}$ is the learned classification function, i.e. the value $\boldsymbol{F}_{vc}$ gives us the (unnormalized) probability that node $v$ belongs to a class $c$, and we can make predictions by taking the argmax. The problem can be solved in closed form (even though in practice one would use power iteration) and the solution is: $\boldsymbol{F} = (1 - \alpha) \left( \boldsymbol{I}_N - \alpha \boldsymbol{D}^{-\sigma} \boldsymbol{A} \boldsymbol{D}^{\sigma-1} \right)^{-1} \boldsymbol{H}$ for $\alpha = 2/(2 + \mu)$. We can see that setting $\sigma = 1$, i.e. the standard Laplacian variant [53] we obtain:

$$\boldsymbol{F} = (1 - \alpha) \left( \boldsymbol{I}_N - \alpha \boldsymbol{D}^{-1} \boldsymbol{A} \right)^{-1} \boldsymbol{H} = \boldsymbol{\Pi} \boldsymbol{H} \tag{7}$$

From Eq. 7 we have that Label Propagation is very similar to our $\boldsymbol{\pi}$-PPNP: instead of diffusing logits which come from a neural network it propagates the one-hot vectors of the labeled nodes instead. From here onwards we apply our proposed method without any modifications by simply providing a different $\boldsymbol{H}$ matrix in Problem 1.

We can also certify the feature propagation (FP) approach of which there are several variants: e.g. the normalized Laplacian FP [7], or a recently proposed equivalent model termed simple graph convolution (SGC) [47]. Feature propagation is carried out in two steps: (i) the node features are diffused to incorporate the graph structure $\boldsymbol{X}^{\text{diff}} = \boldsymbol{\Pi} \boldsymbol{X}$, and (ii) a simple logistic regression model is trained using the diffused features $\boldsymbol{X}^{\text{diff}}$ and subset of labelled nodes. Now, let the $\boldsymbol{W} \in \mathbb{R}^{D \times K}$ be the weights corresponding to a trained logistic regression model. The predictions for all nodes are calculated as $\boldsymbol{Y} = \text{softmax} = (\boldsymbol{X}^{\text{diff}} \boldsymbol{W}) = \text{softmax}(\boldsymbol{\Pi} \boldsymbol{X} \boldsymbol{W}) = \text{softmax}(\boldsymbol{\Pi} \boldsymbol{H})$ with $\boldsymbol{H} = \boldsymbol{X} \boldsymbol{W}$. Thus, again by simply providing a different matrix $\boldsymbol{H}$ in Problem 1 we can certify feature propagation.

## 8.2 Further experiments

In Fig. 5a we show the percent of certifiable robust nodes for different local budgets on the Pubmed graph ($N = 19,717, |\mathcal{E}| = 44,324, D = 500, K = 3$) [37] demonstrating that our method scales to large graphs. Similar to before (Fig. 3a), the models are more robust to attackers that can only remove edges. In Fig. 5b we analyze the robustness of Citeseer w.r.t. increasing global budget. The global budget constraints are again able to successfully restrict the attacker. The global budget makes a larger difference when the attacker has a larger local attack strength ($s = 10$). In Fig. 5c we show

(a) Pubmed, local budget    (b) Citeseer, global budget    (c) Citeseer, robust training

Figure 5: (a,b) The local and global budget successfully restrict the attacker. Models are more robust to removing edges than both removing and adding edges. (c) Our robust training successfully increases the percentage of certifiably robust nodes.

(a) Citeseer, certifiable accuracy.  (b) Runtime: SBM, local (VI).  (c) Runtime: SBM, global (RLT).

Figure 6: Further experiments on certifiable accuracy (a) and runtime (b-c).

that the robust training increases the percent of certifiably robust nodes. Comparing to Fig. 4c we conclude that training with a larger local attack strength ($s = 10$ as opposed to $s = 6$) makes the model more robust overall while the predictive performance ($F_1$ score) is the same in both cases.

We also investigate certifiable accuracy. The ratio of nodes that are both certifiably robust and at the same time have a correct prediction is a lower bound on the overall worst-case classification accuracy since the worst-case perturbation can be different for each node. We plot this ratio in Fig. 6a for Citeseer and see that the certifiable accuracy is relatively close to the clean accuracy when the budget is restrictive, and it decreases gracefully as we in increase the budget.

To show how the runtime scales with number of nodes and number of edges we randomly generate SBM graphs of increasing size, and we set all edges in the generated graphs as fragile ($\mathcal{F} = \mathcal{E}$). In Fig. 6b we see the mean runtime across five runs for local budget (VI algorithm). Even for graphs with more than 10K nodes the certificate runs in a few seconds. Similarly, Fig. 6c shows the runtime for global budget (RLT relaxation). We see that the runtime scales linearly with the number of edges. Furthermore, the overall runtime can be easily reduced by: (i) stopping early whenever the worst-case margin becomes negative, (ii) using Gurobi's distributed optimization capabilities to reduce solve times, and (iii) having a single shared preprocessing step for all nodes.

## 8.3 Proofs

*Proof. Proposition 1.* Problem 2 can be formulated as an average cost infinite horizon Markov decision problem, where at each node $v$ we decide which subset of $\mathcal{F}_v$ edges are active, i.e. $\mathcal{A}_v = \mathcal{P}(\mathcal{F}^v)$ where $\mathcal{P}(\mathcal{F}^v)$ is the power set of $\mathcal{F}^v$ and the reward depends only on the starting state but not on the action and the ending state $r(v, a) = \boldsymbol{r}_v, \forall a \in \mathcal{A}_v$. From the average cost infinite horizon optimality criterion as shown by Fercoq et al. [18] we have:

$$\lim_{T \to \infty} \frac{1}{T} \mathbb{E}\Big( \sum_{t=0}^{T-1} r\big(X_t, \nu_t\big) \Big) = \lim_{T \to \infty} \frac{1}{T} \mathbb{E}\Big( \sum_{t=0}^{T-1} r_{X_t, j} \overline{\nu}_j\big(X_t\big) \Big) = \sum_{i,j \in [n]} \boldsymbol{\pi}_i \boldsymbol{P}_{i,j} r_{i,j} \qquad (8)$$

where $X_t \in \mathcal{S}$ is a random variable denoting the state of the system at the discrete time $t \geq 0$, and $\nu(h_t)$ is deterministic control strategy determining a sequence of actions and is a function of the history $h_t = (X_0, \nu_0, \dots, X_{t-1}, \nu_{t-1}, X_t)$. For this problem there exists a stationary (feedback) strategy $\overline{\nu}(X_t)$ that does not depend on the history such that for all $t \geq 0, \nu_t(h_t) = \overline{\nu}(X_t)$. Eq. 8 follows from the ergodic theorem for Markov chains. Here the reward is more general and can be set depending on the edge $(i, j)$. Letting $r_{ij} = \boldsymbol{r}_i, \forall j$ and plugging it in Eq. 8 we get that the optimality criterion equals $\boldsymbol{r}^T \boldsymbol{\pi}$ since the trasion matrix $\boldsymbol{P} = \boldsymbol{D}^{-1} \boldsymbol{A}$ is row-stochastic. As shown by Hollanders et al. [25] policy iteration is well suited to optimize PageRank and our Algorithm 1 corresponds to policy iteration with local budget for the above MDP. For a fixed damping factor $\alpha$ (which is our case) policy iteration always converges in less iterations than value iteration [34] and does so in weakly polynomial time that depends on the number of fragile edges [25].

□

*Proof. Proposition 2.* Eqs. 4b and 4c correspond to the LP of the unconstrained MDP on the auxiliary graph. Intuitively, the variable $x_v$ maps to the PageRank score of node $v$, and from the variables $x_{ij}^0 / x_{ij}^1$ we can recover the optimal policy: if the variable $x_{ij}^0$ (respectively $x_{ij}^1$) is non-zero then in the

optimal policy the fragile edge $(i, j)$ is turned off (respectively on). Since there exists a deterministic optimal policy, only one of them is non-zero but never both. Eq. 4d corresponds to the local budget. Remarkably, despite the variables $x_{ij}^0/x_{ij}^1$ not being integral, since they share the factor $\frac{x_i}{d_i}$ from Eq. 4c we can exactly count the number of edges that are turned off or on using only linear constraints. Eqs. 4e and 4f enforce the global budget. From Eq. 4e we have that whenever $x_{ij}^0$ is nonzero it follows that $\beta_{ij}^1 = 0$ and $\beta_{ij}^0 = 1$ since that is the only configuration that satisfies the constraints (similarly for $x_{ij}^1$). Intuitively, this effectively makes the $\beta_{ij}^0/\beta_{ij}^1$ variables "counters" and thus, we can utilize them in Eq. 4f to enforce the total number of perturbed edges to not exceed $B$.

We also have to show that solving the MDP on the auxiliary graph solves the same problem as the MDP on the original graph. Recall that whenever we traverse any edge from node $i$ we obtain reward $\boldsymbol{r}_i$. On the other hand, whenever we traverse an edge from the auxiliary node $v_{ij}$ corresponding to a fragile edge $(i, j)$ to the node $i$ (action "off") we get negative reward $-\boldsymbol{r}_i$, and the transition probability is 1. Intuitively, traversing back and forth between node $i$ and node $v_{ij}$ does not change the overall reward obtained (since $\boldsymbol{r}_i$ and $-\boldsymbol{r}_i$ cancel out). That is, we have the same reward as in the original graph with the edge $(i, j)$ excluded. Similarly, when we traverse the edge from auxiliary node $v_{ij}$ to the node $j$ (action "on") we obtain 0 reward, i.e. no additional reward is gained and the transition happens with probability $\alpha$. Therefore, the overall reward is the same as if the fragile edge $(i, j)$ would be present in the original graph.

More formally, for any given arbitrary policy for the unconstrained MDP on the auxiliary graph, let $k_v$ be the current number of "off" fragile edges for node $v$ and let $\mathcal{F}_+^v$ be the current set of "on" fragile edges. From Eqs.4b and Eqs.4c we have:

$$x_v - \alpha \sum_{(i,v) \in \mathcal{E}_f \cup \mathcal{F}_+^v} x_i d_i^{-1} - k_v d_v^{-1} = (1-\alpha)\boldsymbol{z}_v \qquad (9a)$$

$$x_v = \alpha \sum_{(i,v) \in \mathcal{E}_f \cup \mathcal{F}_+^v} x_i d_i^{-1} + (1-\alpha)\boldsymbol{z}_v - k_v d_v^{-1} \implies x_v = \boldsymbol{\pi}(\boldsymbol{z}_v)_v - k_v d_v^{-1} \qquad (9b)$$

where we can see that $\boldsymbol{\pi}(\boldsymbol{z}_v)_v$ is the personalized PageRank for node $v$ for a perturbed original graph corresponding to the current policy, i.e. the graph where all $(v, j) \in \mathcal{F}_+^v$ for all $v \in \mathcal{V}$ are turned "on". Plugging in Eq. 9b into the objective from Eq. 4a we have

$$\max \sum_{v \in \mathcal{V}} x_v \boldsymbol{r}_v - \sum_{(i,j) \in \mathcal{F}} x_{ij}^0 \boldsymbol{r}_i = \max \sum_{v \in \mathcal{V}} \boldsymbol{\pi}(\boldsymbol{z}_v)\boldsymbol{r}_v$$

which exactly corresponds to the objective of Problem 2. Since the above analysis holds for any policy it also holds for the optimal policy, and therefore solving the unconstrained MDP on the auxiliary graph is equivalent to solving the unconstrained MDP on the original graph.

Combining everything together we have that solving the QCLP is equivalent to solving Problem 2.

$\square$

*Proof. Proposition 3.* Using the reformulation-linearization technique (RLT) we relax the quadratic constraints in Eq. 4e. In general, from RLT it follows that we add the following four linear constraints for each pairwise quadratic constraint $m_i m_j = M_{ij}$

$$M_{ij} - \underline{m}_i m_j - \underline{m}_j m_i \geq -\underline{m}_i \underline{m}_j \qquad (10a)$$

$$M_{ij} - \underline{m}_j m_i - \overline{m}_i m_j \leq -\underline{m}_j \overline{m}_i \qquad (10b)$$

$$M_{ij} - \underline{m}_i m_j - \overline{m}_j m_i \leq -\underline{m}_i \overline{m}_j \qquad (10c)$$

$$M_{ij} - \overline{m}_i m_j - \overline{m}_j m_i \geq -\overline{m}_i \overline{m}_j \qquad (10d)$$

where $\underline{m}_i \leq m_i \leq \overline{m}_i$ are lower and upper bounds for $m_i$.

From Eq. 4e we see that our quadratic terms always equal to 0 ($M_{ij} = 0$), and we have the following upper $\overline{\beta_{ij}^0} = \overline{\beta_{ij}^1} = 1$, and $\overline{x_{ij}^1} = \overline{x_{ij}^0} = \frac{x_i}{d_i} > 0$, and lower bounds $\underline{\beta_{ij}^0} = \underline{\beta_{ij}^1} = \underline{x_{ij}^1} = \underline{x_{ij}^0} = 0$. Plugging these upper/lower bounds into Eq. 10 for our quadratic terms $\overline{x_{ij}^0 \beta_{ij}^1} = 0$ and $\overline{x_{ij}^1 \beta_{ij}^0} = 0$ we see that the constraints arising from Eqs. 10a, 10b and 10c are always trivially fulfilled. Thus we are left with the constraints arising from Eq. 10d which for our problem are:

$$x_{ij}^0 + \overline{x_{ij}^0} \beta_{ij}^1 \leq \overline{x_{ij}^0} \qquad \text{and} \qquad x_{ij}^1 + \overline{x_{ij}^1} \beta_{ij}^0 \leq \overline{x_{ij}^1} \qquad (11)$$

There are two cases to consider:

Case 1: The edge is turned "off". We have $x_{ij}^0 = x_i d_i^{-1}$ and $x_{ij}^1 = 0$.

$$x_{ij}^0 + \overline{x_{ij}^0}\beta_{ij}^1 \leq \overline{x_{ij}^0} \implies x_{ij}^0(\overline{x_{ij}^0})^{-1} + \beta_{ij}^1 \leq 1 \implies$$
$$\implies x_{ij}^0(\overline{x_{ij}^0})^{-1} \leq \beta_{ij}^0 \implies x_{ij}^0(\overline{x_i}d_i^{app_proofs-1})^{-1} \leq \beta_{ij}^0$$

And trivially: $x_{ij}^1 + \overline{x_{ij}^1}\beta_{ij}^0 \leq \overline{x_{ij}^1} \implies \overline{x_{ij}^1}\beta_{ij}^0 \leq \overline{x_{ij}^1} \implies \beta_{ij}^0 \leq 1$.

Case 2: The edge is turned "on". We have $x_{ij}^1 = x_i d_i^{-1}$ and $x_{ij}^0 = 0$.

$$x_{ij}^1 + \overline{x_{ij}^1}\beta_{ij}^0 \leq \overline{x_{ij}^1} \implies x_{ij}^1(\overline{x_{ij}^1})^{-1} + \beta_{ij}^0 \leq 1 \implies x_{ij}^1(\overline{x_i}d_i^{-1})^{-1} \leq \beta_{ij}^1$$

And trivially: $\overline{x_{ij}^0} + \overline{x_{ij}^0}\beta_{ij}^1 \leq \overline{x_{ij}^0} \implies \overline{x_{ij}^0}\beta_{ij}^1 \leq \overline{x_{ij}^0} \implies \beta_{ij}^1 \leq 1$

The above two cases are disjoint and we can plug $\beta_{ij}^0$ and $\beta_{ij}^1$ into Eq. 4f to obtain Eq.5. $\qquad\square$

## 8.4 SDP relaxation

In this section we show that the SDP-relaxation [43] based on semidefinite programming is not suitable for our problem since the constraints are trivially fulfilled. For convinience, we rename the variables that participate in the quadratic constraints $(\beta_{ij}^0, x_{ij}^0, \dots)$ to $(y_1, y_2, \dots)$. The SDP relaxation replaces the product terms $y_i y_j$ (e.g. $x_{ij}^0\beta_{ij}^1$) by an element $\boldsymbol{Y}_{ij}$ of an $n \times n$ matrix $\boldsymbol{Y}$ and adds the constraint $\boldsymbol{Y} - \boldsymbol{y}\boldsymbol{y}^T \succeq 0$, where $\boldsymbol{y}$ is the vector of variables. Since in the original $QCLP$ there are no terms of the form $y_i y_i$ corresponding to the elements on the diagonal, we can make the diagonal elements $\boldsymbol{Y}_{ii}$ arbitrarily high to make the matrix $\boldsymbol{Y} - \boldsymbol{y}\boldsymbol{y}^T$ positive semidefinite and trivially satisfy the constraint.

## 8.5 Hardness of PageRank optimization with global budget

The Link Building problem [32, 33] aims at maximizing the PageRank of a single given node $v$ by selecting a set of $k$ optimal edges that point to node $v$. We will use the fact that the Link Building problem is a special case of Problem 2 to derive our hardness result.

**Problem 3** (Link Building [32]). *Given a graph $G = (\mathcal{V}, \mathcal{E})$, node $v \in \mathcal{V}$, budget $k \in \mathbb{Z}$, and any fixed $\alpha \in (0, 1)$. Find a set $\mathcal{S} \subseteq \mathcal{V} \setminus \{v\}$ with $|S| = k$ maximizing $\boldsymbol{\pi}_{\tilde{G},\alpha}(\boldsymbol{e}/n)_v$ in the perturbed graph $\tilde{G} = (\mathcal{V}, \tilde{\mathcal{E}} := \mathcal{E}_f \cup (\mathcal{S} \times \{v\}))$, where $\boldsymbol{e}/n$ is the teleport vector for the uniform distribution.*

**Proposition 4.** *Problem 2 with global budget is W[1]-hard and allows no FPTAS.*

*Proof.* Setting the teleport vector to the uniform distribution $\boldsymbol{z} = \boldsymbol{e}/n$, the reward vector to $\boldsymbol{r} = \boldsymbol{e}_v$, the set of fragile edges to $\mathcal{F} = (\mathcal{V} \setminus \{v\}) \times \{v\}$, the set of fixed edges to $\mathcal{E}_f = \mathcal{E}$, and configuring the budgets as $b_v = 1, \forall v$ and $B = k$ we see that the Problem 3 is a special case of Problem 2. Note that, since we can always increase $\boldsymbol{\pi}_v$ by adding edges pointing to $v$, the $x \leq B$ global constraint is equivalent to the $x = B$ constraint where $x$ is the expression on the left-hand side in Eq. 4f.

Olsen [32] shows that the Link Building problem is W[1]-hard and admits no FPTAS by reducing it to the Regular Independent Set problem which is W[1]-complete [8]. Therefore, Problem 2 with global budget is also W[1]-hard and allows no FPTAS since $k$ is preserved in the reduction. $\qquad\square$

## 8.6 Alternative upper bound

As an alternative upper bound for $x_v$ we can use the following approach: Assume we have given a fixed set of edges $\mathcal{E}_f$ where every node has at least one fixed edge. From Proposition 2 we have $x_v = (1 - k_v d_v^{-1})^{-1}\boldsymbol{\pi}_v$. To maximize this value, we can simply set $\boldsymbol{\pi}(\boldsymbol{z})_v = 1$ (since this is the maximal PageRank score achievable) and $k_v = |\mathcal{F}^v|$. Since every node has at least one fixed edge, we have $d_v > k_v$, i.e. the inverse is always defined.

## 8.7 Further experimental details

We preprocess each graph and keep only the nodes that belong to largest connected component. The resulting graph for Cora-ML has $N = 2,810$, $|\mathcal{E}| = 10,138$, for Citeseer $N = 2,110$, $|\mathcal{E}| = 7,336$ and for Pubmed $N = 19,717$, $|\mathcal{E}| = 88,648$. Unless otherwise specified we set $\alpha = 0.85$. We compute the certificates with respect to the predicted class label, i.e. we set $y_t$ in $m^*_{y_t,*}(t)$ to the predicted class for node $t$ using the clean graph. Experiments are run on Nvidia 1080Ti GPUs using CUDA and TensorFlow and on Intel CPUs. We use the GUROBI solver to solve the linear programs.

We configure our $\pi$-PPNP model with one hidden layer and choose a latent dimensionality of 64. We randomly select 20 nodes per class for the training/validation set, and use the rest for the testing. The weights $\theta$ are regularized with the $L_2$ norm with strength of $5e - 2$. We train for a maximum of $10,000$ epoch with a fixed learning rate of $1e - 2$ and patience of 100 epochs for early stopping. We train the model for five different random splits and report the averaged results.

When reporting results for local budget (e.g. Figs. 3, 4c, 5a, 5c) we evaluate the certifiable robustness for all test nodes, since as we discussed in Sec. 4.3 we only need to run Algorithm 1 $K \times K$ times to obtain certificates for all nodes. When reporting results for global budget (e.g. Figs. 4a and 5b) we randomly select 150 test nodes for which we compute the certificate. For all results regarding runtime (e.g. Figs. 4b, 6b, 6c) we report average time across five runs on a machine with 20 CPU cores.