[Reviews · NeurIPS 2019]

Reviewer 1



Post-rebuttal: Thanks for the response. Would be great to include the rebuttal discussion in the final version of this work. >>>>> This paper proposes a new framework for certified robustness of graph neural networks. The task under consideration is node classification and the threat model allows for addition and removal of edges under local (node-specific) and global upper bounds on the permissible perturbations. Under this threat model, the paper derives efficient algorithms for obtaining robustness certificates using a suitably designed MDP. The margin formulation can further be used to derive robust losses for training. The problem is well-motivated and significant, the writing is clear for most parts, and the theoretical contributions are excellent! The robust training algorithm and experimental validation further lends support to the proposed framework. I haven't checked the proofs in great detail, but overall I enjoyed reading the paper and believe it is a significant contribution to the field. Some comments/questions for the authors: - The theoretical exposition crucially hinges on the fact that the GNN logits for the pre-final layer (and consequently the margin) is linear in the propagating vector (e.g., PageRank scores). This is not true for most GNNs used (e.g., GATs, GCNs). Could the authors shed light on the difficulties in analyzing robustness of non-linear propagation schemes under similar threat models? - The experiments for robust training are done in a transductive setting, i.e., when the full graph is visible. How would the robust training results look for inductive semi-supervised node classification (test graph contains more nodes and edges than train graph)?

Reviewer 2



The authors develop a framework for adversarial robustness in graph neural networks where the adversary can change a budgeted number of edges. The adversary is constrained such that they cannot change more than a budgeted number of edges as well as a certain number of edges from the same node. The authors then want to certify the robustness of the graph neural network from robustness under this threat model of [Klicpera et al., 2019]. This model uses the "predict then propagate" framework: node features are learned in isolation then they are "propagated" by weighting the local features by the personalized PageRank. Thus the graph structure is used in a simpler way than other graph neural networks. When there is only the "local" budget, the authors develop an exact algorithm. Their approach is based on Markov decision processes similar to (Fercoq et al., 2010). When there is a global budget, the authors write the problem as a quadratic program and relax the problem using the "Reformulation Linearization Technique". They show that many constraints in the relaxed problem can be simplified to a single linear inequality. Finally they experimentally verify their methods. They show that certain aspects improve robustness, such as increasing the teleport probability in personal PageRank. The authors mention that [Klicpera et al., 2019] do not use personalized PageRank, however I believe that they do. To improve clarity, the authors should move the description of the reformulation linearization technique in the appendix to the main paper. The authors write "the SDP-relaxation [30] is not suitable for our problem since the constraints are trivially fulfilled". It would be interesting and helpful to explain why. How long does the method based on the reformulation linearization technique take to run? Further, how is the certificate quality when there are no local constraints, only global constraints? Is it possible to show that adding the global constraints makes the problem NP-hard? It might be possible to also solve this problem optimally. *** Post Rebuttal *** Thank you for including the NP-hardness reduction in the rebuttal.

Reviewer 3



Originality: The problem the paper study is innovative and interesting. However, the techniques the paper relies are variation and combination of existing methods. Quality: The paper is technically sound with proofs for all the claims. The paper is reproducible as the authors provide code and detailed description of experiments. Clarity: The paper is well written and very easy to follow. Significance: The applicability of the method is kind of limited. The method and analysis are limited to a very specific type of GNN. On the contrary, the most widely applied GNN method are based on message-passing and convolution like GraphSage. The techniques in this paper is very hard to generalize beyond the PPNP model.

[Author Response · NeurIPS 2019]

**Certifiable Robustness to Graph Perturbations: Author Response**

**R1/R2/R3: Limited Focus.** As suggested, we will clarify in the paper that our focus is on certifying PPNP and label/feature propagation; and not every possible GNN. Certifying any of these approaches is highly relevant: e.g. label propagation is quite popular in practice (often as part of more complicated pipelines in industry), and the strong empirical performance of PPNP has already been independently verified [1]. We can also trivially extend our approach to certify a recently proposed model termed Simple Graph Convolution (SGC) [2] which is equivalent to feature propagation. Certifying SGC is useful since it is one of the few GNNs that demonstrates scalability to graphs with millions of nodes. In future work, we can extend our approach to GCNs by using a similar analysis to Xu et al. [3] (Theorem 1) which shows that the influence between nodes in a k-layer GCN is proportional to a k-step random walk distribution by e.g. bounding the influence with (truncated) PageRank to obtain a certificate.

**R2: SDP relaxation.** Let $(y_1, y_2, \dots)$ be the variables corresponding to $\beta_{ij}^0, x_{ij}^0$, etc. The SDP relaxation replaces the product terms $y_i y_j$ by an element $Y_{ij}$ of an $n \times n$ matrix $Y$ and adds the constraint $Y - yy^T \succeq 0$. Since in the original $QCLP$ there are no terms of the form $y_i y_i$ corresponding to the elements on the diagonal, we can make the diagonal elements $Y_{ii}$ arbitrarily high to make the matrix $Y - yy^T$ positive semidefinite and trivially satisfy the constraint.

**R2: NP-hard proof.** We provide a proof sketch that adding the global budget makes the problem NP-hard by constructing a polynomial reduction from the 1-IN-3SAT problem which is NP-complete. The problem: Given a boolean 3-CNF formula s.t. the clauses contain only un-negated atoms, does there exist a truth assignment s.t. in each clause, exactly one literal is true. First, add a single node $t$, and one node for each literal $l_1, \dots, l_n$ and each clause $c_1, \dots, c_m$. Let $\mathcal{E}_f$ (non-fragile set) contain: one edge from each node to $t$, one edge from $t$ to each clause $c_j$, and one edge from each clause $c_j$ to its three literals ($3m$ in total). Let $\mathcal{F}$ (fragile set) contain $3m$ edges, one from each literal $l_i$ to its clauses, and let $\mathcal{E} = \mathcal{E}_f \cup \mathcal{F}$. Set the global budget $B = 2m$, and the teleport vector and reward vector as $\mathbf{z} = \mathbf{r} = \mathbf{e}_t$. Such reward means that we are maximizing the PageRank $\pi(\mathbf{z})_t$ of the single node $t$, or equivalently minimizing the expected first hitting time $h_t$ to $t$. Intuitively, for this graph removing any fragile edge decreases $h_t$, which means we can always improve the objective by removing more edges, up to the budget $B = 2m$. Thus, there are exactly $m$ fragile edges left (i.e. $2m$ removed) in the optimal configuration $\mathcal{O}^*$. Let $f_j$ be the number of fragile edges in $\mathcal{O}^*$ pointing to clause $c_j$. Claim: 1-IN-3SAT is satisfiable iff in the *optimal* solution each $f_j = 1$. First note that for any optimal solution, if one edge from some literal is in $\mathcal{O}^*$ then all edges from that literal are in $\mathcal{O}^*$ (up to the budget). The reason is that adding an additional edge from a literal already in $\mathcal{O}^*$ to some clause leads to a smaller $h_t$ increase than adding an edge from a literal not yet in $\mathcal{O}^*$ to some clause. Given this, the right-to-left direction of the claim above is trivial: Since each $f_j = 1$, every clause has exactly one literal set to true. It follows: 1-IN-3SAT is satisfiable. Left-to-right: Given that 1-IN-3SAT is satisfiable. Assume that the optimal configuration $\mathcal{O}'$ contains some clause $c_1$ with $f_1 = 2$. Since $|\mathcal{O}'| = m$, there must be a clause $c_2 \neq c_1$ with $f_2 = 0$. Now $c_1$ forms 2 cycles with its literals which increases $h_t$, but having $f_2 = 0$ decreases $h_t$. The former increase is always larger then the later decrease, thus a configuration where some $c_j$'s have $f_j = 2$ always has a larger $h_t$ compared to any configuration where all $f_j = 1$. Since such a configuration exists (satisfiability holds), $\mathcal{O}'$ cannot be optimal. Similarly, this holds if some $f_j = 3$. Thus, it follows that if 1-IN-3SAT is satisfiable $\mathcal{O}^*$ recovers the truth assignment and all $f_j = 1$.

**R2: Only global budget.** Our approach is not designed to handle only global budget since the proposed upper bounds explicitly depend on having local budget. Deriving tight upper bounds for the "global only" case is left for future work.

**R2/R3: Runtime.** To show how the runtime scales with number of nodes we randomly generate SBM graphs of increasing size. In Fig. 1a we see the mean runtime for local budget (VI algorithm). Even for graphs with more than 10K nodes the certificate runs in a few seconds. Similarly, Fig. 1b shows the runtime for global budget (RLT relaxation). The runtime can be easily reduced by: (i) stopping early whenever the worst-case margin becomes negative, (ii) using Gurobi's distributed optimization capabilities to reduce solve times, and (iii) having single preprocessing for all nodes.

**R3: Overall accuracy.** Notice that the ratio of nodes that are both certifiably robust and at the same time have a correct prediction is a lower bound on the overall classification accuracy since the worst-case perturbation can be different for each node. We plot this ratio in Fig. 1c for Citeseer. We will include this finding in the updated paper.

[1] Fey, M. and Lenssen, J. E. Fast graph representation learning with pytorch geometric. *arXiv:1903.02428*, 2019.
[2] Wu et al. Simplifying graph convolutional networks. In *ICML 2019*.
[3] Xu, K. et al. Representation learning on graphs with jumping knowledge networks. In *ICML 2018*.

(a) Runtime: local (VI).

(b) Runtime: global (RLT).

(c) Bound on certifiable accuracy.

[Meta-Review · NeurIPS 2019]

The paper presents a framework to certify the robustness of graph neural networks where the adversary has a limited budget for changing edges in the network. All three reviewers found the work interesting, significant, and of high quality. The author response was highly responsive to reviewer comments and included a sketch of a requested NP-completeness proof. The AC recommends accept.